

# From sea ice to seals: A moored marine ecosystem observatory in the Arctic

Claudine Hauri[1], Seth Danielson[2], Andrew M. P. McDonnell[2], Russell R. Hopcroft[2], Peter Winsor[2], Peter Shipton[2], Catherine Lalande[3], Kathleen M. Stafford[4], John K. Horne[5], Lee W. Cooper[6], Jacqueline M. Grebmeier[6], Andrew Mahoney[7], Klara Maisch[8], Molly McCammon[9], Hank Statscewich[2], Andy Sybrandy[10], Thomas Weingartner[2]

[1]International Arctic Research Center, University of Alaska Fairbanks, Fairbanks, AK, 99775, USA
[2]College of Fisheries and Ocean Science, University of Alaska Fairbanks, Fairbanks, AK, 99775, USA
[3]Département de Biologie, Université Laval, Québec, QC, G1V 0A6, Canada
[4]Applied Physics Laboratory, University of Washington, Seattle, WA, 98195, USA
[5]School of Aquatic and Fishery Sciences, University of Washington, Seattle, WA, 98195, USA
[6]Chesapeake Biological Laboratory, University of Maryland Center for Environmental Science, Solomons, MD, 20688, USA
[7]Geophysical Institute, University of Alaska Fairbanks, Fairbanks, AK, 99775, USA
[8]Klara Maisch Art and Design, Fairbanks, AK, 99775, USA
[9]Alaska Ocean Observing System, Anchorage, AK, 99501, USA
[10]Pacific Gyre Inc., Oceanside, CA, 92056, USA

Correspondence to: Claudine Hauri (chauri@alaska.edu)

**Abstract.** Although Arctic marine ecosystems are changing rapidly, year-round monitoring is currently very limited and presents multiple challenges unique to this region. The Chukchi Ecosystem Observatory (CEO) described here uses new sensor technologies to meet needs for continuous, high resolution, and year-round observations across all levels of the ecosystem in the biologically productive and seasonally ice-covered Chukchi Sea off the northwest coast of Alaska. This mooring array records a broad suite of parameters that facilitate observations, yielding better understanding of physical, chemical and biological couplings, phenologies, and the overall state of this Arctic shelf marine ecosystem. While cold temperatures and eight months of sea ice cover present challenging conditions for the operation of the CEO, this extreme environment also serves as a rigorous test bed for innovative ecosystem monitoring strategies. Here, we present data from the 2015-16 CEO deployments that provide new perspectives on the seasonal evolution of sea ice, water column structure and physical properties, annual cycles in nitrate, dissolved oxygen, phytoplankton blooms and export, zooplankton abundance and vertical migration, the occurrence of Arctic cod, and vocalizations of marine mammals such as bearded seals. These integrated ecosystem observations are being combined with ship-based observations and modeling to produce a time-series that documents biological community responses to changing seasonal sea ice and water temperatures while establishing a scientific basis for ecosystem management.

## 1 The Gateway to the Arctic Ocean

The Chukchi continental shelf is the seasonally ice-covered entryway of Pacific-origin waters flowing northward into the Arctic Ocean. An oceanic pressure and elevation differential between the Pacific and the Arctic Oceans is the driving force for this transport (Stigebrandt, 1984), moving water, heat, nutrients, organic carbon, and organisms northward, leading to transformations on the shelf en route to the deep Arctic Ocean. Large late spring and summertime phytoplankton blooms (Sambrotto et al., 1984; Springer et al., 1996; Arrigo et al., 2014; Hill et al., in press) make the Chukchi continental shelf an extremely productive marine ecosystem that supports a thriving benthos (Grebmeier et al., 2006), zooplankton (Ershova et al., 2015), seabirds (Kuletz et al., 2015) and marine mammals (Hannay et al., 2013).

The Chukchi Sea shelf is part of a broader Arctic system undergoing rapid change. The Arctic near surface air temperature is increasing almost twice as fast as the global average (Serreze and Francis, 2006; Stocker et al., 2013). On the





Chukchi shelf, annual average temperatures have been as much as 0.8 °C warmer during the last two decades compared to the average of the 1900-2016 period of record (Smith et al., 2008, Fig. 1). Warming has led to a > 40% Arctic-wide decrease of summertime sea ice extent over the last 4 decades (Serreze and Stroeve, 2015). In the Chukchi and Beaufort seas, ice cover has decreased by 1.24 days/year since 1979, a trend that accelerated to a decrease of 12.84 days/year in the 2000-2012 period (Frey et al., 2015). The freshwater content of the Arctic Ocean has also increased profoundly since the 1990's, with potentially large effects on the global thermohaline circulation (McPhee et al., 2009; Proshutinsky et al., 2009). Anti-cyclonic winds, sea ice melt, increased precipitation and run-off are suggested to be the contributing factors to the widespread freshening of the Arctic Ocean (McPhee et al., 2009; Morison et al., 2012; Bintanja and Selten, 2014). Pacific Arctic storm frequency and intensity has also increased over the last 25 years (Pickart et al., 2013). This increased storm activity corresponds with wintertime Northern hemisphere temperature increases, which has likely led to a northward shift of Northern Hemisphere storm tracks (McCabe et al., 2001; Hakkinen et al., 2008). High latitude marine ecosystems are also particularly vulnerable to ocean acidification (Orr, 2011). Due to naturally lower carbonate ion concentrations $[CO_3^{2-}]$ and accelerated decrease of $[CO_3^{2-}]$ as a result of sea ice and glacial melt (Yamamoto-Kawai et al., 2009; Evans et al., 2014), these regions are quickly being pushed closer or past biologically important thresholds. Already today, the consequences of these anthropogenic changes are visible in the marine ecosystem and manifest themselves as species range shifts, changes in abundance, growth, condition, behaviour and phenology, and community and regime shifts (Wassmann et al., 2011).

These anthropogenic changes have large implications for the ecosystem, and the global carbon cycle and climate. To monitor these changes, disentangle their effects from those caused by natural variability, and improve our mechanistic understanding of the ecosystem dynamics, we designed an observatory capable of continuously recording a broad suite of ecosystem parameters in the northeastern Chukchi Sea (Fig. 2-4).

## 2 The Chukchi Ecosystem Observatory

The CEO is an array of closely co-located subsurface moorings in the Northeast Chukchi Sea (71° 35.976' N 161° 31.621' W, Fig. 2-4). The CEO is situated in 45 m of water on the southeastern flank of Hanna Shoal within a productive biological "hotspot" (Grebmeier et al., 2015). The shoal's shallow depths result in deep ice keel groundings (Barrett and Stringer, 1978), and accumulation of thick ice, which serves as important habitat for walrus and other animals (Jay et al., 2012). The exact observatory siting and our ecological understanding of the greater region is based on many years of multi-disciplinary sampling on the NE Chukchi shelf, including those of the Chukchi Sea Environmental Studies Program (Day et al., 2013), the Chukchi Offshore Monitoring in Drilling Area (Dunton et al. 2014), and the Distributed Biological Observatory (DBO; Moore and Grebmeier, 2018).

The CEO moorings carry sensors that collectively measure an extensive suite of physical, biogeochemical, and biological parameters (Fig. 4). These sensors allow us to observe and understand the phenology and connections within this Arctic marine ecosystem. The sensors capture temporal variations in sea ice cover and thickness, light, currents, waves, water column structure, and concentrations of dissolved oxygen, nitrate, inorganic carbon species, and particulate matter. They document the presence of phytoplankton blooms and export, zooplankton abundance and vertical migration, the presence of Arctic cod and other fishes, and the vocalizations of marine mammals. The CEO is designed to monitor the ecosystem year-round, making it well-suited for studying interactions among ecosystem components, especially during the poorly documented winter months. Although to our knowledge, no other single Arctic monitoring site measures the full suite of parameters collected by the CEO, of course many of the individual measurements are also made elsewhere across the Arctic. With time, we expect that insights derived from the CEO observations will be extended to other Arctic shelf ecosystems and trigger new comparative studies.

## 3 The Chukchi Seascape

The artist's depiction of the Hanna Shoal ecosystem (Fig. 3) illustrates the seasonal cycle at the CEO site (Fig. 3A). Moving from left to right, Fig. 3 captures seasonal shifts from the ice-covered winter, into the productive summer, and finally into the stormy and biologically senescent autumn.





Through heat loss, sea ice formation, and brine rejection (Fig. 3B) in late fall and winter, the water column over the Chukchi shelf becomes more saline and vertically homogenized (Weingartner et al., 2005). Although nutrients are in abundant supply from the incoming Anadyr-origin waters, planktonic production remains limited due to the scarcity of light (Fig. 3C). Over the course of winter, continued heat loss to the atmosphere leads to thermodynamic thickening of the sea ice,

while convergence of the ice pack leads to mechanical thickening in the form of pressure ridges. Divergence of the ice pack creates open water in the form of leads and polynyas, which during winter will freeze and thicken through the same processes. As the light begins to return in the spring, diatoms and other algae begin to bloom within the ice matrix and at the ice-water interface (Fig. 3D; Ambrose et al., 2005; Gradinger, 2009).

It is not until late May or June, when the days are long, insolation is strong, and warm water moves in from the

south that sea ice begins to melt, thin, and recede northwards (Fig. 3E). During this time, the water column stratifies with inputs of fresh meltwater and heat at the surface (Fig. 3G) and extraordinary phytoplankton blooms occur in the nutrient rich surface waters (Fig. 3F; Hill et al., in press). These processes set up strong vertical gradients of inorganic carbon and nutrients across the shallow water column (Fig. 3H; Bates, 2006). Combined with relatively low grazing activity, the high rates of primary production support large fluxes of sinking particulate organic matter to the sea floor (Fig. 3I, Lalande et al.,

2007), thereby sustaining a rich benthic ecosystem (Fig. 3J; Grebmeier et al., 2015), which attracts large numbers of marine mammals that forage on the benthos (Fig. 3K; Jay et al., 2012; Hannay et al., 2013) or Arctic cod (Fig. 3L).

Fall is characterized by surface cooling and more frequent and intense storm systems with strong winds (Fig. 3M) that erode the highly stratified water column. This process brings remineralized nutrients and inorganic carbon from bottom waters into the surface layer, supporting modest fall phytoplankton blooms and the outgassing of carbon dioxide into the

atmosphere (Else et al., 2012; Hauri et al., 2013). Later, as sunlight fades into the darkness of winter, primary production further slows, the planktonic ecosystem becomes senescent (Fig. 3N), and the benthos continues to thrive off of organic matter stored in the sediments (Pirtle-Levy et al., 2008).

Due to the logistical complexities of operating in the region, most of the observational work done in the Pacific sector of the Arctic Ocean takes place during the sea ice-free summer and early autumn via research vessels (Fig. 3O).

Autonomous vehicles such as gliders have also found increasing use in recent years (Fig. 3P; Baumgartner et al., 2014; Martini et al., 2016; Danielson et al., 2017).

**4 Arctic Observing Challenges**

A starting premise of our effort to improve understanding of this complex ecosystem and monitor ongoing changes, is that it is necessary to extend observations of the ecosystem into the ice-covered winter and employ new observational

approaches that are appropriate for this challenging environment.

Given the presence of deep ice keels that regularly exceed 20 m depth - and may occasionally extend as deep as 30 m - we restrict the uppermost sensor package of our observatory to 33 m below the surface, leaving only 12 m of the water column safe for mooring instrumentation and hardware. Although we cannot deploy instruments in the upper 30 m of the water column when sea ice is present, upward looking acoustic instruments in the array provide observations above the top-

mooring package. An Acoustic Zooplankton Fish Profiler (AZFP, manufactured by ASL Environmental Sciences) measures the presence and abundance of zooplankton and fish and ice draft, while a TeledyneRDI Acoustic Doppler Current Profiler (ADCP) records current velocity and direction. During sea ice free conditions, the ADCP instrument also quantifies the height, period and direction of surface waves.

One example of a purpose-built technology for the CEO is a novel "freeze-up detection mooring" that was first

deployed in fall 2015 (Fig. 4c). Oceanographers have long struggled with finding a way to measure upper water column stratification and heat content through the fall up to the time of freeze-up in ice-covered seas. The freeze-up detection mooring was outfitted with an expendable surface float that housed a satellite communications package, a tether release, an inductive modem, and a sea surface temperature sensor. The surface float was connected to four Sea-Bird SBE 37 inductive modem CTDs that transmitted hourly temperature, salinity and pressure to the surface float from four subsurface depths (8,

20, 30, and 40 m), along with a sub-surface camera that records and sends digital images of the upper water column. The advance of the fall ice pack was closely monitored with satellite imagery and the surface float provided simultaneous real-time monitoring of the temperature and salinity throughout the water column leading up to ice formation. When the ice edge





was within one day of over-running the mooring and sparse ice chunks were already floating by, the surface float was remotely released from the mooring, leaving a mid-depth subsurface float to provide floatation for the portion left behind. The data from this mooring are presented and discussed below (Fig. 5).

Cold seawater (temperatures below 0 °C for most of the year) decreases the capacity of all batteries. Some instruments are powered with lithium batteries that provide a higher power density. Engineering constraints dictate the trade-offs between the various sensor battery packs and the desired sampling rates. For example, due to the large power demand of the Kongsberg Contros HydroC pCO$_2$ sensor, its sampling rate had to be decreased to once every 24 hours but the AZFP instrument has sufficient power and memory to sample every 15 seconds for the entire year.

A dedicated vessel charter for servicing the remote CEO is also not cost-effective given the 2000 km distance to the nearest deep water, year-round ice-free port Dutch Harbor in Unalaska, Alaska. We thereby rely on vessels of opportunity during the summer months to deploy and recover the CEO as part of other funded shipboard research. By partnering with other oceanographic research teams that are operating in the region, we are also able to collect water samples from the CEO site to provide for in situ calibration of sensors throughout the deployment. Furthermore, ship-based observational efforts such as the Arctic Marine Biodiversity Observatory Network and the DBO programs add spatial context to the CEO data.

Conversely, the CEO can help place research cruise data into a fuller temporal context, including variability on scales ranging from the synoptic to the inter-annual (Danielson et al., 2017).

Despite design advances and features, limitations and challenges still remain to be overcome. For example, without real-time data communications capabilities, instrument function and data returns can only be assessed annually following the CEO turn-around. This delays data availability and makes the approach poorly suited for adaptive sampling efforts after

deployment. Furthermore, throughout the winter, there are currently no ship-based efforts or autonomous vehicles operating in the region. The result is limited spatial context during winter and early spring, and an inability to collect samples for frequent calibrations of the CEO's deployed sensors. Many of the measurements made from the CEO are collected at a single depth within the water column, thus limiting interpretation of upper water column parameters, especially over the winter months. Future innovations and investments in technologies such as profiling winches, direct-to-shore submarine

cable communications, or under-ice autonomous assets are a few possibilities that could mitigate some of these challenges.

## 5 First Scientific Results

The 2015-16 data returns from the CEO provide a unique window into the year-round Arctic marine ecosystem (Fig. 6). We also present new data showing water column turnover and cooling processes during the freeze up period (Fig. 5 and 6).

The physical conditions measured at the CEO include currents, waves, temperature, salinity, ice draft and light (Photosynthetic Active Radiation, PAR) (Fig. 5 and 6a-d). The temperature and salinity cycles are tied to lateral advection, surface heat fluxes, ice cover and winds. Sensors on the freeze-up detection buoy show that the upper 20 m of the water column was well mixed and steadily lost heat from around 4 °C at the beginning of September to -1.5 °C shortly before sea ice formed at the beginning of November (Fig. 5). Water temperature at 30 m depth largely followed bottom water (40 m)

temperature during September and early October. However, several large departures when 20 and 30 m temperatures briefly warmed and even exceeded those at the surface could indicate the effects of lateral advection and/or the passing influence of intrapycnal eddies (Lu et al., 2015). The bottom water temperature steadily increased from -1.5 °C at the end of September to 1 °C during freeze up at the beginning of November, reversing the vertical temperature gradient in mid October. Between September and November wind speeds in excess of 10 m s$^{-1}$ were observed during the passage of several low-pressure

systems (Fig. 5a). At peak intensity, these storm events did not appear to erode stratification at the CEO site. However wind direction reversals, from predominantly upwelling-favorable directions (northeasterly) to downwelling-favorable directions (southwesterly), were associated with periodic depressions of the pycnocline. After freeze-up, water temperatures at the 34 and 43 m depths remained near the freezing point (-1.6 to -1.8 °C) through the end of the record in August.

Salinity followed a more cyclical progression of freshening between June and November and salinization in the

other half of the year. Ice cover persisted for nearly nine months (November through August) and the thermodynamic thickening and thinning of the ice can be seen in the overall shape of the ice draft time-series (Fig. 6a). Some ice keels extended to deeper than 10 m below the surface, although an ice draft of 1-3 meters was more typical. The absence of deeper



keels may be due to the upwind proximity of Hanna Shoal, which would likely block or deflect deep-keeled ridges from the Northeast. At the same time, the absence of extended mid-winter periods of open water demonstrates that the CEO lies outside any polynya formation zone due to the proximity of the shallow Hanna Shoal.

Nitrate is the limiting nutrient in the Chukchi Sea (Walsh et al., 1989) and therefore is an important bottom-up control for the ecosystem. We deployed a SUNA V2 nitrate sensor (Sea-Bird Scientific) on the upper instrument package (34 m below the surface, Fig. 6e). Over 26-28 August 2015, shortly after deployment, nitrate values dropped from above 15 µM down to between 5 and 7.5 µM during a strong and prolonged storm. Nearby shipboard wind measurements exceeded 10 m s$^{-1}$ for part of each day from August 25-29, 2015. The drop in nitrate concentrations was simultaneous with a sharp increase in bottom water temperatures (Fig. 6c) and an increase in dissolved oxygen (Fig. 6f), likely indicating that strong mixing of

surface waters with warm, nutrient-depleted and oxygen-rich waters down to the depth of the sensors. Nitrate and temperature remained relatively constant for a couple of weeks before returning to higher and lower values, respectively, in the middle of September. While dissolved oxygen concentrations increased slowly from mid-September until freeze up, nitrate concentrations declined as water column stratification weakened and overturning was initiated (Fig. 5) due to the strong heat and buoyancy losses from the surface ocean. These coincident changes indicate that the decline in nitrate during

this time period was driven in part by the dilution of bottom water nitrate with low-nitrate surface waters, although some fall production also occurred at this time: chlorophyll *a* fluorescence at the CEO was measurable through at least early November (Fig. 6g). Following freeze-up, nitrate concentrations slowly increased from a low of approximately 5 µM in early November to 12 µM in early May. These increases and the anti-correlated decrease in dissolved oxygen reflect the ongoing remineralization of organic matter in sediments and the water column throughout the winter.

Chlorophyll *a* fluorescence and sediment trap collections reveal a seasonal cycle of phytoplankton and ice algae export from surface waters. Large chlorophyll *a* peaks were observed in August and September. In June, chlorophyll *a* fluorescence increased from a low wintertime background level and remained elevated with intermittent peaks throughout the summer. Because the sensor is located below the summer mixed layer, it is not well situated to record the phytoplankton that reside within the upper water column or within/under the ice. However, the identification of sediment trap contents

indicates that *Nitzschia frigida*, an ice-obligate pennate diatom, began sinking from the ice as early as April, with large pulses in May and June (Fig. 6h). During these times, sea ice and surface snowmelt would have begun, a process that would flush diatoms out of the ice matrix. Large quantities of phytoplankton were also collected in the traps in September/October and June/July, correlating with the chlorophyll *a* peaks, suggesting blooms in spring and fall.

Wavelet analysis of the AZFP acoustic backscatter at 125 kHZ indicated a strong diurnal (1 day period) signal (Fig.

6j). This diurnal signal was found in open water conditions in fall when the zooplankton undergo daily migrations up and down in the water column. More surprisingly, there were also strong indications of diurnal migrations in mid-winter (January to March) under ice-cover. In the spring, backscatter at 125 kHz was present but not strongly associated with diurnal migration.

The 38 kHz active acoustic data suggests a strong diurnal migration of fish with swim bladders from August into

October, diminished backscatter from December to February, and then higher background levels with intermittently strong returns in April through July (Fig. 6k).

The passive acoustic spectra reveal the timing and source of underwater sounds, notably bearded seal vocalizations in spring and early summer (Fig. 6l).

## 6 Importance of the Observatory

Over the past several decades, moorings have been deployed at select sites across the Arctic as tools for providing insights into the year-round functioning of this system (e.g. Woodgate et al., 2012; Nishino et al., 2016; Polyakov et al., 2017; de Jong et al., 2018). These moorings are usually outfitted with sensors to measure physical properties and, less frequently, biological and geochemical samplers, such as nitrate sensors and sediment traps. To obtain a better understanding of polar ecosystems, ship-based biological and geochemical sampling often complement moored observatories (e.g. Southern

Ocean Observing System, Fram Strait Arctic Outflow Observatory). Much has been learned from these endeavours, especially with regard to water mass circulation and the functioning of the ecosystem and biological pump (e.g. Forest et al., 2013, Kenitz et al., 2017; Erikson et al., 2018). The CEO adds a highly outfitted complement of acoustic, optical,



electrochemical, and gas membrane sensors as well as direct sample collection devices. In doing so, the CEO dataset illuminates multiple linkages of the physical, chemical, and biological environment. This broad suite of observations represents a necessary but mostly untried approach to integrated ecosystem research and monitoring in the Arctic. To our knowledge, the CEO is the most extensive moored observatory for continuous recording of ecosystem parameters in an ice-
covered sea.

     The Arctic has already, and will continue to undergo transformative physical and chemical changes. Such changes may trigger a cascade of consequences that propagate into the regional ecosystem and may test its resiliency and vulnerability. The extensive year-round dataset derived from the CEO is providing insights into how the ecosystem operates from physics to biology. These baseline data also offer quantitative comparisons in future years for assessing ecosystem
responses to an altered climate. Even though distinction of secular trends from natural interannual, decadal, and seasonal variability will require a time-series of around 40 years in high latitude regions (Henson et al., 2010), gaining a better understanding of the system's current state and mechanisms that govern its variability are necessary first steps towards that goal. The data and improved mechanistic understanding of the shelf ecosystem are available to improve biogeochemical and ecological models that allow us to test, analyze, and prepare for the future. The status and trends in the marine ecosystem of
the northeastern Chukchi Sea, observed through seasonal ship field programs, moorings such as the CEO, and satellite observations will provide critical information on the status of the ecosystem and associated ecosystem services it can provide. For example, local subsistence users are interested in a healthy foodweb that supports traditional food sources. Reduced sea ice may increase the potential of northward migration of subarctic species, including commercial fish species that will alter those foodwebs. The CEO will allow year-round tracking of the marine ecosystem in the northeastern Chukchi
Sea, and thus can provide data valuable to an ecosystem-based approach to resource management.

**7 Concluding Thoughts**

     Arctic changes, including human induced influences on climate, can be expected to affect high latitude food webs. A better understanding of the driving factors of potential ecosystem shifts can only be gained through coordinated and simultaneous measurements such as these that span a wide range of physical, chemical and biological indicators. Ocean
acidification, warming, freshening, and de-oxygenation are large-scale issues that require interdisciplinary efforts. While ship based multidisciplinary efforts remain valuable components of observing efforts, these only provide episodic data coverage. On the other hand, continuous time-series moorings generally do not include such a large array of disciplines (Newton et al., 2015). The model of extensive ecosystem observation capabilities that the CEO provides can be used in other ecosystems beyond the challenging Arctic environment. For example, an ecosystem observatory for the Gulf of Alaska in the
subarctic Pacific is currently being developed and is using some of the lessons learned here.

**Author contribution**
     SD developed and designed the CEO. CH, SD, AMPM, RRH, PW, PS, CL, KMS, HS, and JKH maintain instrumentation on the observatory. PS built and deployed the observatory. KM created the artwork in collaboration with
CH, SD and AMPM.  CH prepared the manuscript with contributions from all co-authors.

**Competing interests**
     The authors declare that they have no conflict of interest.

**Acknowledgements**
     A broad consortium of academic, agency, and industry partners are contributing to the success of the CEO program. Major financial support comes from the National Pacific Research Board Long-term Monitoring (NPRB LTM) program (project #1426), the Alaska Ocean Observing System (award #NA11NOS0120020), and the University of Alaska Fairbanks. Additional partners include the University of Washington, Université Laval, Olgoonik-Fairweather, LLC. Claudine Hauri
acknowledges support from the National Science Foundation Office of Polar Programs (OPP-1603116). Projects that serviced the CEO and/or collected water column calibration data were funded by the National Science Foundation, Bureau of Ocean Energy Management, National Oceanic and Atmospheric Administration, National Oceanographic Partnership



Program, and Shell Exploration and Production Company, Inc. The primary goal of the NPRB LTM is to support time-series research and thereby improve the predictability of ecosystem responses to changing ocean conditions. Maintenance and calibration of our remote observatory is only possible due to numerous collaborators within the Arctic research community who helped with CEO deployment and recovery or collected sensor calibration samples. We would therefore like to thank
Carin Ashjian, Jessica Cross, Miguel Goñi, Burke Hales, Katrin Iken, Laurie Juranek, Calvin Mordy, and Robert Pickart. Post-processed data are available through http://www.chukchiecosystemobservatory.org/ and http://www.aoos.org. Data from the ice detection buoy is available at: https://portal.aoos.org/arctic#metadata/75373/station. This is NPRB publication #XXXX (to be added)

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





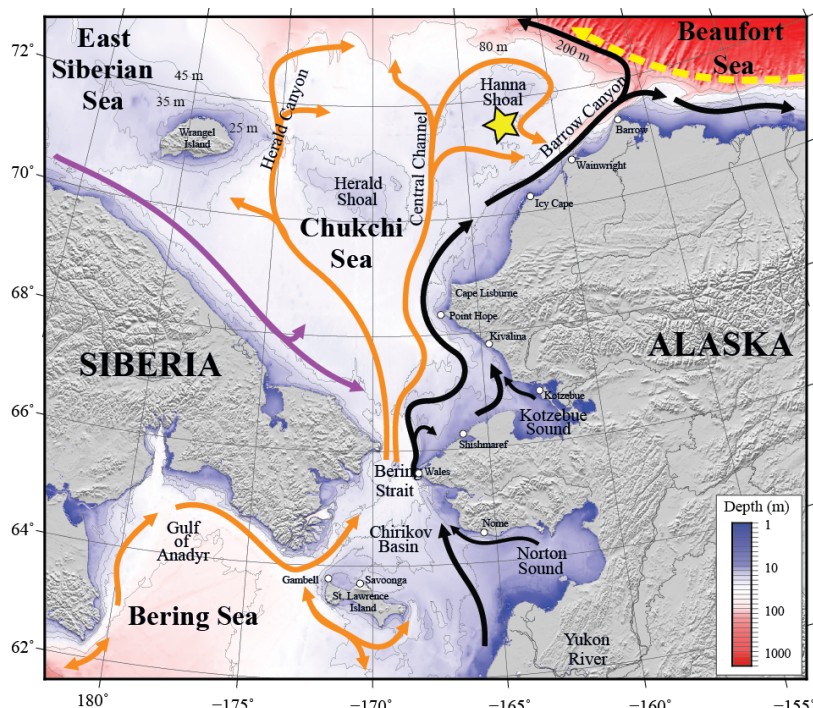

**Figure 2: Bathymetry of the Chukchi, northern Bering, East Siberian and eastern Beaufort seas. The Chukchi Ecosystem Observatory (CEO) near Hanna Shoal is marked with a yellow star. General circulation patterns are shown with arrows: Black: Alaskan Coastal Water and Alaskan Coastal Current, dividing into the Shelfbreak Jet (right) and Chukchi Slope Current (left, Corlett and Pickart, 2017); Orange: Anadyr, Bering Sea, and Chukchi Sea Water; Purple: Siberian Coastal Current; Yellow: Beaufort Gyre boundary current.**



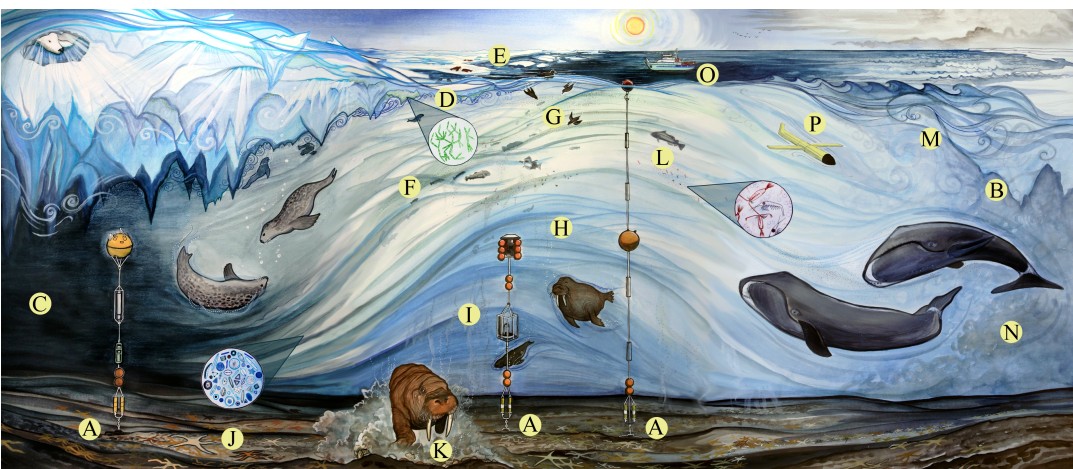

5 **Figure 3: Illustration of the mooring array and the ecosystem at the observatory site. A) Mooring array, B) brine rejection, C) dark winter, D) sea ice algae bloom, E) receding sea ice, F) phytoplankton bloom, G) stratification, H) vertical gradient of nutrients and inorganic carbon, I) sinking particulate organic matter, J) rich benthic ecosystem, K) foraging walrus, L) Arctic cod, M) storm-induced mixing, N) senescent planktonic ecosystem, O) Research Vessel Sikuliaq, and P) glider. Examples of walrus and bearded seal sounds are available at: http://mather.sfos.uaf.edu/~seth/CEO/Sounds.html.**





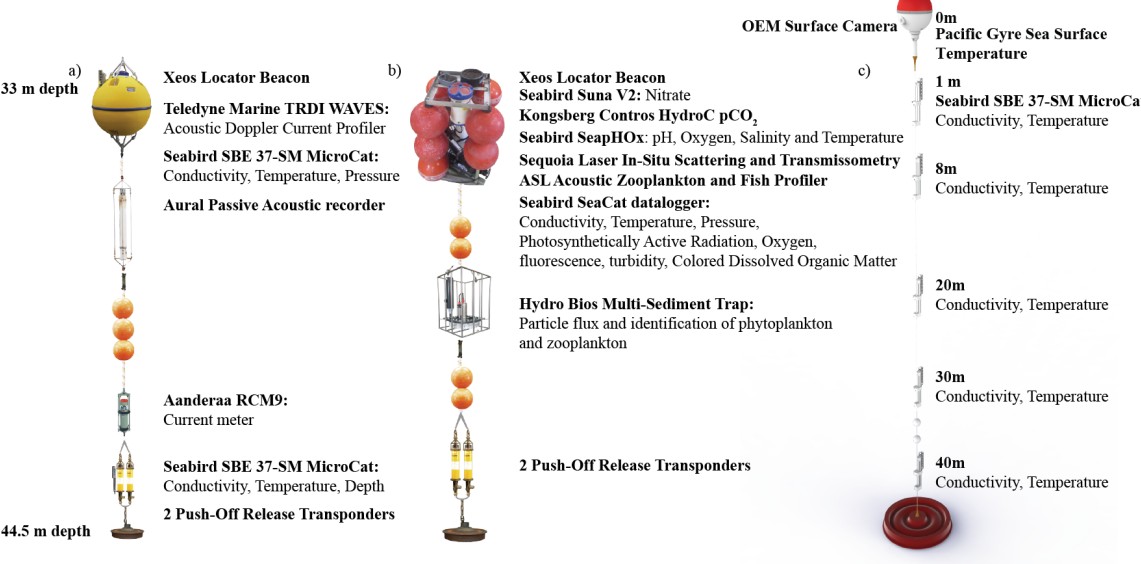

**Figure 4: Illustration of the Chukchi Ecosystem Observatory (CEO), including the a) biophysical, b) biogeochemical, and c) freeze-up moorings. Note that the freeze up mooring includes a surface package, whereas the other two moorings only reach up to 33 m depth. Sensors are calibrated with as many *in-situ* samples as possible. Calibration samples are always collected upon deployment and recovery of the moorings, and depending on other research activity nearby our site, also at other times of the year. $pCO_2$ and pH sensors were added in summer 2016.**



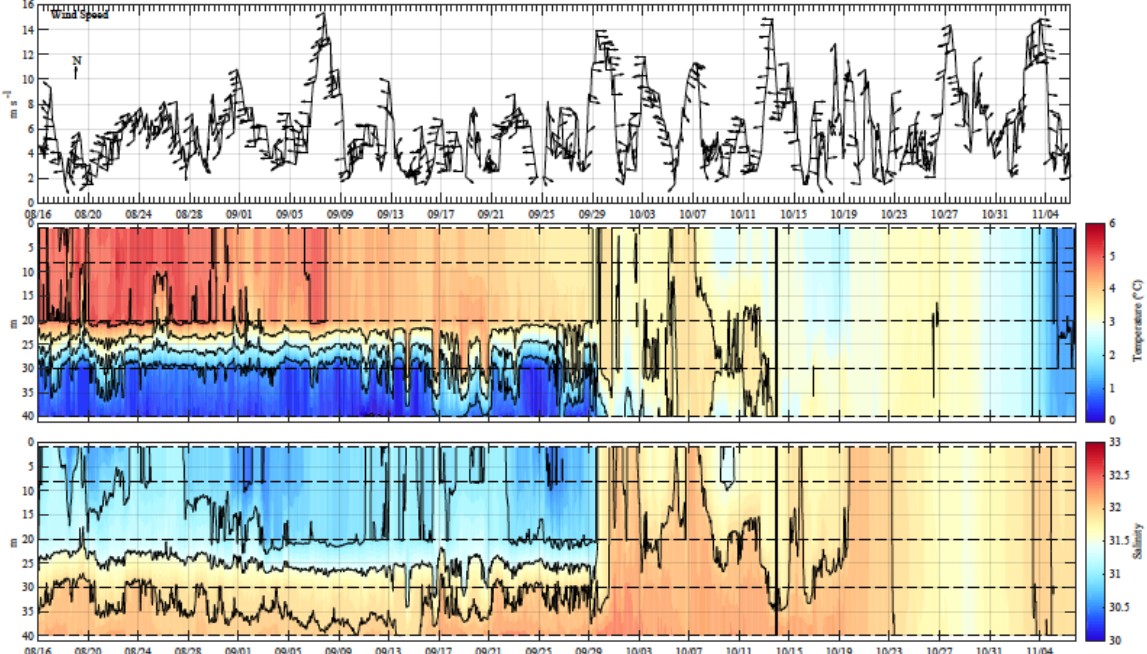

Figure 5: Time series from the NOAA operated Barrow Airport weather station and the Chukchi Ecosystem Observatory freeze-up detection mooring deployment in 2015. Shown are a) wind speed and direction (arrows pointing downwind), b) temperature (°C), and c) salinity. Conductivity and temperature sensors were moored at 8 m, 20 m, 30 m, and 40 m (dashed lines) from early September until close to freeze up at the beginning of November.



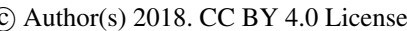

**Figure 6: Data from the 2015/2016 deployment. Shown are a) ice draft (m), b) salinity, c) temperature (°C), d) photosynthetically active radiation (PAR, uE cm$^{-2}$ s$^{-1}$), e) nitrate (NO$_3$, umol l$^{-1}$), f) oxygen (O$_2$, umol kg$^{-1}$), g) fluorescence (mg m$^{-3}$), *Nitzschia frigida* flux (million cells m$^{-2}$ d$^{-1}$), i) Diatoms (million cells m$^{-2}$ d$^{-1}$) j) acoustic zooplankton fish profiler (AZFP, days) 125 KHz, k) AZFP 38 KHz (days), and l) acoustic spectra (Hz). *In-situ* NO$_3$ water samples were collected at times of the CEO deployment and recovery, and were analysed with standard wet chemical determinations of nitrate + nitrite of frozen samples at the Chesapeake Biological Laboratory. Using the calibration samples as anchor points, a drift of 12 umol l$^{-1}$ throughout the deployment was found and corrected by linearly detrending the data.**