# Peer review of "From sea ice to seals: A moored marine ecosystem observatory in the Arctic"

_Ocean Science, 2018_

## Referee Comment (RC1) · D. R. Eriksen (Referee) · 3 Sep 2018

General comments Thankyou for the opportunity to review this paper. The current state of research and new approaches described are an important contribution to improving high frequency time-series in locations that are traditionally hard to access for year-round observations. The observatory described, located in a region of dynamic change, is impressive, and although this paper presents preliminary analysis of results, the compilation of information on how the observatory was designed and factors that resulted in the final design are incredibly important for other research programs that use complex moorings arrays in harsh environments. The value of communicating "lessons learned" cannot be understated. I look forward to the following series of papers that provide in depth analysis of these high frequency observations and improved under-

standing of ecosystem dynamics in this region. Minor comments: P 2, l 42. Figure 4. Because this is such a great illustration of a time-series of change in a complex environment, I immediately went looking for the artists name, it took me a little while to find it. I wonder if you can highlight this more in the caption. Or provide a download link to a high -quality version that can be used with appropriate citation and acknowledgement. P3, l 10 Do you have continuous data for estimates of MLD at the mooring location? Even a simple summary of how this changes relative to water column depth over the seasonal cycle would be useful for those of us more familiar with Antarctic cycles than Arctic cycles and ocean dynamics. P 3, l 13- can you provide a citation for the "relatively low grazing activity"? This is an interesting point for understanding modes of carbon export compared to other polar systems. Also, any linkages to zooplankton phenology associated with both the summer and fall phytoplankton blooms. P 3, l39 this freeze-up mooring and associated data set is fantastic. I can see wide applications. P 4 l 12 Can you comment on how many times during a typical mooring deployment cycle you were able to obtain samples for sensor calibration purposes? The option of profiling winches is certainly attractive option, with the potential for "event based "sampling if real-time communications limitations can be overcome P 6 l 45 SOTS observatory is described by "Eriksen et al" (not "Erikson")

---

## Referee Comment (RC2) · Anonymous Referee #2 · 9 Oct 2018

The authors present a blueprint for a complete observing system, capable of capturing relevant physical, chemical and biological parameters for monitoring and understanding the progress of change in a vulnerable Arctic system.

While the paper doesn't really focus on new scientific results, the manuscript is a useful way to share the challenges and early successes of working with an automated observing platform in a remote region. I hope that papers with a deeper focus on the science are forthcoming.

The description of the physical environment and the well-studied physical seasonality provides a nice background for the description of the observing platform and the presentation of preliminary results.

[Figure]

The freeze up detection method sounds great – was the 'expendable float' recovered? Would be a tough sell to leave gear like that behind by design in the Antarctic . . ...

Minor Comments:

It would have been nice to see the CO2 system data that was collected once the SeapHOx sensor was added since these provide the data required to address questions relevant to the progress of acidification which the authors outline as a key research goal.

It was reassuring to see the correspondence in temperature at 34 and 43 meters depth since measurements are restricted to the subsurface – any ideas about seasonality of mixed layer depth? The pCO2 supersaturations observed In Canadian waters in winter were more shallow than 34 meters (see also Shadwick et al., 2011 L&O).

---

## Author Comment (AC1) · 20 Oct 2018

(1) Comments from Referee 1

Comment by D. R. Eriksen (Referee) ruth.eriksen@csiro.au

General comments: Thank you for the opportunity to review this paper. The current state of research and new approaches described are an important contribution to improving high frequency time-series in locations that are traditionally hard to access for year-round observations. The observatory described, located in a region of dynamic change, is impressive, and although this paper presents preliminary analysis of results, the compilation of information on how the observatory was designed and factors

that resulted in the final design are incredibly important for other research programs that use complex moorings arrays in harsh environments. The value of communicating "lessons learned" cannot be understated. I look forward to the following series of papers that provide in depth analysis of these high frequency observations and improved understanding of ecosystem dynamics in this region.

Minor comments: P 2, l 42. Figure 4. Because this is such a great illustration of a time-series of change in a complex environment, I immediately went looking for the artists name, it took me a little while to find it. I wonder if you can highlight this more in the caption. Or provide a download link to a high -quality version that can be used with appropriate citation and acknowledgement.

P3, l 10 Do you have continuous data for estimates of MLD at the mooring location? Even a simple summary of how this changes relative to water column depth over the seasonal cycle would be useful for those of us more familiar with Antarctic cycles than Arctic cycles and ocean dynamics.

P 3, l 13- can you provide a citation for the "relatively low grazing activity"? This is an interesting point for understanding modes of carbon export compared to other polar systems. Also, any linkages to zooplankton phenology associated with both the summer and fall phytoplankton blooms.

P 3, l39 this freeze-up mooring and associated data set is fantastic. I can see wide applications.

P 4 l 12 Can you comment on how many times during a typical mooring deployment cycle you were able to obtain samples for sensor calibration purposes? The option of profiling winches is certainly attractive option, with the potential for "event based "sampling if real-time communications limitations can be overcome

P 6 l 45 SOTS observatory is described by "Eriksen et al" (not "Erikson")

(2) Author's response

Dear Dr. Eriksen, Thank you very much for taking the time to review our manuscript. We are currently working on a series of papers describing our data in more detail and are hoping to be able to submit these papers soon.

We really appreciate your comments about the art and are hoping that it will be used widely. Besides highlighting Klara Maisch's name in the caption, we will also provide a link to a high-resolution version of her art that will be hosted on her private website, including an appropriate citation.

We unfortunately do not have seasonal data for the mixed layer depth at the mooring location. The ice detection buoy is a first try to get a better understanding of the mixed layer depth from the time it is deployed (late July or August) until freeze up. Peralta-Ferriz and Woodgate, 2015 (Progress in Oceanography, now referenced in paper) compiled available salinity and temperature profiles from 1979-2012 and estimated a mixed layer depth minimum of 12 m in July/August and maximum of 36 m in March. Current methods to collect seasonal mixed layer depth data are too costly for our project (e.g. WHOI's bottom lander), but we are looking into other options.

There are several studies that suggest a relatively low grazing rate for the Chukchi Sea. For example, Campbell et al 2009 (Campbell, R.G., E.B. Sherr, C.J. Ashjian, S. Plourde, B.F. Sherr, V. Hill, and D.A. Stockwell. 2009. Mesozooplankton prey preference and grazing impact in the western Arctic Ocean. Deep Sea Research Part II: Topical Studies in Oceanography 56 (17): 1274-1289) and Kitamura M, Amakasu K, Kikuchi T, Nishino S (2017) Seasonal dynamics of zooplankton in the southern Chukchi Sea revealed from acoustic backscattering strength. Continental Shelf Research 133:47-58 doi:https://doi.org/10.1016/j.csr.2016.12.009. However, co-author Catherine Lalande has been working up data from CEO's sediment trap, which contain a lot of fecal pellets, pointing towards high grazing pressure during spring. Her hypothesis is that even if there is a productive zooplankton community, primary production is so extremely high and the shelf so shallow that carbon export is considerable. These results and zooplankton phenology associated with the summer and fall blooms, including a discussion of papers reporting low grazing pressure will be presented in a manuscript that is currently in preparation.

We always take calibration samples at deployment and recovery of the observatory. Unfortunately, since the HydroC pCO2 and SeapHOx sensors need some acclimatization time ($\sim$ 2 weeks for SeapHOx), samples taken right after their deployment cannot be used for calibration. However, we are usually able to get 1 to 3 additional calibration samples in fall, depending on cruises of opportunity and the willingness of the PIs to make the effort of collecting water samples.

All comments are directly addressed in the manuscript and described in section (3) "Authors changes in manuscript."

Thank you again for reviewing our manuscript and for your productive comments.

Best regards, Claudine Hauri and co-authors

(3) Author's changes in manuscript

P 2, l 42. Figure 4. Because this is such a great illustration of a time-series of change in a complex environment, I immediately went looking for the artists name, it took me a little while to find it. I wonder if you can highlight this more in the caption. Or provide a download link to a high -quality version that can be used with appropriate citation and acknowledgement. -> We added the following sentences to the caption of figure 3: The illustration was painted by Klara Maisch. A high-resolution version can be downloaded from her personal website at: https://klaramaisch.com/chukchi-sea-mooring-illustration.

P3, l 10 Do you have continuous data for estimates of MLD at the mooring location? Even a simple summary of how this changes relative to water column depth over the seasonal cycle would be useful for those of us more familiar with Antarctic cycles than Arctic cycles and ocean dynamics. -> We added available information about the mixed layer depth to the text. These estimates are based on available data from the entire

Chukchi Sea. We changed the text accordingly: p. 3 L.1: Through heat loss, sea ice formation, and brine rejection (Fig. 3B) in late fall and winter, the water column over the Chukchi shelf becomes more saline and vertically homogenized (Weingartner et al., 2005), deepening the mixed layer depth to a maximum of ~36 m in March (Peralta-Ferriz and Woodgate, 2015). P. 3 L. 9: During this time, the water column stratifies with inputs of fresh meltwater and heat at the surface (Fig. 3G), leading to a shoaling of the mixed layer depth to a minimum of ~12 m (Peralta-Ferriz and Woodgate, 2015). This is the time when extraordinary phytoplankton blooms occur in the nutrient rich surface waters (Fig. 3F; Hill et al., 2018).

P 3, l 13- can you provide a citation for the "relatively low grazing activity"? This is an interesting point for understanding modes of carbon export compared to other polar systems. Also, any linkages to zooplankton phenology associated with both the summer and fall phytoplankton blooms. -> We modified the text to: P.3 L 12: These high rates of primary production support large fluxes of sinking particulate organic matter to the seafloor (Fig. 3I, Lalande et al., 2007), thereby sustaining a rich benthic ecosystem (Fig. 3J; Grebmeier et al., 2006, Grebmeier et al., 2015), which attracts large numbers of marine mammals that forage on the benthos (Fig. 3K; Jay et al., 2012; Hannay et al., 2013) or Arctic cod (Fig. 3L).

P 3, l39 this freeze-up mooring and associated data set is fantastic. I can see wide applications. -> Thank you!

P 4 l 12 Can you comment on how many times during a typical mooring deployment cycle you were able to obtain samples for sensor calibration purposes? The option of profiling winches is certainly attractive option, with the potential for "event based "sampling if real-time communications limitations can be overcome -> Please see the comment above. To calibrate the 2015-2016 NO3 data record, we used two calibration samples as described in the figure 6 caption, p16, L5-8: "In-situ NO3 water samples were collected at times of the CEO deployment and recovery, and were analysed with standard wet chemical determinations of nitrate + nitrite of frozen samples at the

Chesapeake Biological Laboratory. Using the calibration samples as anchor points, a drift of 12 umol l-1 throughout the deployment was found and corrected by linearly detrending the data."

P 6 l 45 SOTS observatory is described by "Eriksen et al" (not "Erikson") We corrected the typo.

Please also note the supplement to this comment:
https://www.ocean-sci-discuss.net/os-2018-82/os-2018-82-AC1-supplement.pdf

---

## Author Comment (AC2) · 20 Oct 2018

(1) Comments from Referee 1

Anonymous Referee #2

The authors present a blueprint for a complete observing system, capable of capturing relevant physical, chemical and biological parameters for monitoring and understanding the progress of change in a vulnerable Arctic system. While the paper doesn't really focus on new scientific results, the manuscript is a useful way to share the challenges and early successes of working with an automated observing platform in a remote region. I hope that papers with a deeper focus on the science are forthcoming. The description of the physical environment and the well-studied physical seasonality provides a nice background for the description of the observing platform and the presentation of preliminary results.

The freeze up detection method sounds great – was the 'expendable float' recovered? Would be a tough sell to leave gear like that behind by design in the Antarctic.

Minor Comments: It would have been nice to see the CO2 system data that was collected once the SeapHOx sensor was added since these provide the data required to address questions relevant to the progress of acidification which the authors outline as a key research goal. It was reassuring to see the correspondence in temperature at 34 and 43 meters depth since measurements are restricted to the subsurface – any ideas about seasonality of mixed layer depth? The pCO2 supersaturations observed In Canadian waters in winter were more shallow than 34 meters (see also Shadwick et al., 2011 L&O).

(2) Author's response

Dear Referee, We are grateful to you for taking the time to review our manuscript, thank you. We are currently working on a series of papers describing our data in more detail and are hoping to be able to submit these papers soon. The reason we did not include pH and pCO2 data in this current paper is that the timeseries of pH and pCO2 only starts in the summer of 2016. We chose to show the data return of our 2015/2016 deployment because it contained the most complete record of post-processed data to date. Similar to drifting buoys, the top float of our ice freeze up buoy is left behind. It contains the satellite communications modem, and some very basic electronics to trigger the release and a battery, which make up about 1/3 of the price. The vast majority of the cost of the mooring is recovered, including all scientific sensors and acoustic releases. We have put effort into trying to make them more biodegradeable by using fewer plastics and this is an ongoing process.

We unfortunately do not have seasonal data for the mixed layer depth at the mooring location. The ice detection buoy is a first try to get a better understanding of the mixed

layer depth from the time it is deployed (late July or August) until freeze up. Peralta-Ferriz and Woodgate, 2015 (Progress in Oceanography, now referenced in MS) compiled available salinity and temperature profiles from 1979-2012 and estimated a mixed layer depth minimum of 12 m in July/August and maximum of 36 m in March. Please keep in mind that these estimates are based on available data from the entire Chukchi Sea, and may not precisely reflect the seasonality of the mixed layer depth at the observatory location. Current methods, such as using benthic landers to collect seasonal mixed layer depth data are outside the scope of our project, but we are looking into other options. While analyzing our biogeochemical data, we will keep in mind that the mixed layer depth may at times be deeper than the location of our sensors.

(3) Author's changes in manuscript

Any ideas about seasonality of mixed layer depth? -> We added available information about the mixed layer depth to the text. These estimates are based on available data from the entire Chukchi Sea. We changed the text accordingly: p. 3 L.1: Through heat loss, sea ice formation, and brine rejection (Fig. 3B) in late fall and winter, the water column over the Chukchi shelf becomes more saline and vertically homogenized (Weingartner et al., 2005), deepening the mixed layer depth to a maximum of ∼36 m in March (Peralta-Ferriz and Woodgate, 2015).

P. 3 L. 9: During this time, the water column stratifies with inputs of fresh meltwater and heat at the surface (Fig. 3G), leading to a shoaling of the mixed layer depth to a minimum of ∼12 m (Peralta-Ferriz and Woodgate, 2015). This is the time when extraordinary phytoplankton blooms occur in the nutrient rich surface waters (Fig. 3F; Hill et al., 2018).

Please also note the supplement to this comment:
https://www.ocean-sci-discuss.net/os-2018-82/os-2018-82-AC2-supplement.pdf
* * *
[Figure]

**Supplement:**

[revised manuscript text omitted]

---

## Author Response (AR2)

Dear Editor Hoppema,

Thank you very much for thoroughly reading our manuscript again. We appreciate your comments and think that it further improved our manuscript. Please find a detailed description of the changes below.
5 The changes are also visible in the marked-up manuscript.

Thank you again for having taken the time to further edit and improve the paper.

Best regards,
10 Claudine Hauri and co-authors

**Point by point response**

Throughout the manuscript, the thing that is measured is called „parameter". Please consider to use "variable" instead. A parameter should be something that is parameterized, right?
➔ "parameter" has been changed to "variable" throughout the manuscript

Sections 6 and 7 convey a very similar kind of message and for that reason they should be combined into one, i.e. in one section with the title "Concluding remarks" or "Concluding thoughts". The contents can stay as is.
➔ Sections 6 and 7 have been combined
25
P2, L1 „annual average temperatures" It is not clear from the text whether air temperature or sea temperature is meant. Please clarifiy.
➔ It now reads: "annual sea surface temperatures"

30 P3, L42-43 "Oceanographers have long struggled with finding a way to measure upper water column stratification and heat content through the fall up to the time of freeze-up in ice-covered seas." This sentence sounds as if not from a scientific work. Please delete or modify to say what the instrument is supposed to do.
➔ This was changed to:" One example of a purpose-built technology for the CEO is a novel
35 "freeze-up detection mooring", which measures upper water column stratification and heat content through the fall up to the time of freeze-up in ice-covered seas (Fig. 4c). It was first deployed in fall 2015."

P4, L17 I think fuller is not a correct word here, and anywhere else. Something is either full or not, but
40 cannot become fuller. Please change wording.
➔ This was changed to "Conversely, the CEO can help place research cruise data into a broader temporal context,…"

P6, L11 "from physics to biology" I think you are not talking about these disciplines in a strict sense. This is more about physical and biological processes, right? Please modify accordingly.

➔ This was changed to "The extensive year-round dataset derived from the CEO is providing insights into how the ecosystem operates, covering physical, chemical, and biological processes."

P6, L15 "The data and improved mechanistic understanding of the shelf ecosystem are available …" Are these really available now? I think it is too early to speak of improved mechanistic understanding.

➔ We changed it to: "The data from the observatory are available to improve biogeochemical and ecological models that allow us to test, analyze, and prepare for the future."

P6, L20-21 "Reduced sea ice may increase the potential of northward migration of subarctic species, including commercial fish species that will alter those foodwebs." This contention needs a reference.

➔ We added the following reference: Frainer, A., Primicerio, R., Kortsch, S., Aune, M., Dolgov, A.V., Fossheim, M., and Aschan, M. M.: Climate-driven changes in functional biogeography of Arctic marine fish communities, PNAS, 114, 12202-12207, https://doi.org/10.1073/pnas.1706080114, 2017.

P6, L24 "are expected" instead of "can be expected"

➔ Changed!

P6, L26-27 "Ocean acidification, warming, freshening, and de-oxygenation are large-scale issues …" I do not fully agree. Those processes also play on smaller scales, and the variation in different oceanic regions is great.

➔ Quickly arising issues such as ocean acidification, warming, freshening, and de-oxygenation require interdisciplinary efforts.

P6, L29 "do not include such a large array of disciplines" It is not clear what "such" is referring to. I think it must be deleted.

➔ This was changed to "On the other hand, continuous time-series moorings generally do not include a large array of disciplines (Newton et al., 2015)."

References: Please apply the Ocean Science format for all references. Please check to make all references uniform in format and use of abbreviations etc.
P7, L18 Deep-Sea (hyphen)
P7, L25 Add volume and pages: volume 509, pages 479–482
P7, L42 Deep-Sea (hyphen)
P8, L10 (Figure 1) at this place?
P8, L17 Deep-Sea (hyphen)
P8, L26 There is no journal mentioned.
P8, L38 Deep-Sea (hyphen)

P9, L12-13 Correct reference is: McPhee, M. G., A. Proshutinsky, J. H. Morison, M. Steele, and M. B. Alkire (2009), Rapid change in freshwater content of the Arctic Ocean, Geophys. Res. Lett., 36, L10602, doi: 10.1029/2009GL037525.
P9, L30 Deep-Sea Res.

5  P9, L33 Deep-Sea (hyphen)
P10, L15 Deep-Sea (hyphen)
➔ The suggested corrections and additional adjustments have been made to better align with Ocean Science's format for references

10  Caption Fig. 1, L9 delete: respectively
➔ Done
"The annual anomalies were computed by subtracting 0.114 °C." I do not understand why this was done. Please explain.
➔ We clarified this: "The annual anomalies were computed by subtracting the long-term mean,
15  0.114 °C."

Caption Fig.2, L6 Circulation patterns are indicated with "Current" in most cases. However, the orange arrows are called "Water". Please explain or modify.
➔ The flow patterns schematically represent our general understanding but not all regions of the
20  ocean have been bestowed with named currents (and we dont just make them up!). Some regions have had persistent currents identified and named and at these locations we use the term "current".  In other instances currents have not been named and/or the most important characteristic is the type of water flowing by. In these cases we use the Water designations. We added the following explanation: ""
[revised manuscript text omitted]

Microsoft Office …, 10/30/2018 10:16 AM

Microsoft Office …, 10/12/2018 11:56 AM

Lee Cooper's Mac …, 10/16/2018 2:10 PM

Microsoft Office User 10/15/2018 9:35 AM

Microsoft Office User 10/15/2018 9:34 AM

Microsoft Office User 10/15/2018 9:34 AM

Microsoft Office User 10/15/2018 9:36 AM

ecosystem becomes senescent (Fig. 3N), and the benthos continues to thrive off of organic matter stored in the sediments (Pirtle-Levy et al., 2009).

Due to the logistical complexities of operating in the region, most of the observational work done in the Pacific sector of the Arctic Ocean takes place during the sea ice-free summer and early autumn via research vessels (Fig. 3O). Autonomous vehicles such as gliders have also found increasing use in recent years (Fig. 3P; Baumgartner et al., 2014; Martini et al., 2016; Danielson et al., 2017).

**4 Arctic Observing Challenges**

A starting premise of our effort to improve understanding of this complex ecosystem and monitor ongoing changes, is that it is necessary to extend observations of the ecosystem into the ice-covered winter and employ new observational approaches that are appropriate for this challenging environment.

Given the presence of deep ice keels that regularly exceed 20 m depth - and may occasionally extend as deep as 30 m - we restrict the uppermost sensor package of our observatory to 33 m below the surface, leaving only 12 m of the water column safe for mooring instrumentation and hardware. Although we cannot deploy instruments in the upper 30 m of the water column when sea ice is present, upward looking acoustic instruments in the array provide observations above the top-mooring package. An Acoustic Zooplankton Fish Profiler (AZFP, manufactured by ASL Environmental Sciences) measures the presence and abundance of zooplankton and fish and ice draft, while a TeledyneRDI Acoustic Doppler Current Profiler (ADCP) records current velocity and direction. During sea ice free conditions, the ADCP instrument also quantifies the height, period and direction of surface waves.

One example of a purpose-built technology for the CEO is a novel "freeze-up detection mooring", which measures upper water column stratification and heat content through the fall up to the time of freeze-up in ice-covered seas (Fig. 4c). It was first deployed in fall 2015. The freeze-up detection mooring was outfitted with an expendable surface float that housed a satellite communications package, a tether release, an inductive modem, and a sea surface temperature sensor. The surface float was connected to four Sea-Bird SBE 37 inductive modem CTDs that transmitted hourly temperature, salinity and pressure to the surface float from four subsurface depths (8, 20, 30, and 40 m), along with a sub-surface camera that records and sends digital images of the upper water column. The advance of the fall ice pack was closely monitored with satellite imagery and the surface float provided simultaneous real-time monitoring of the temperature and salinity throughout the water column leading up to ice formation. When the ice edge was within one day of over-running the mooring and sparse ice chunks were already floating by, the surface float was remotely released from the mooring, leaving a mid-depth subsurface float to provide floatation for the portion left behind. The data from this mooring are presented and discussed below (Fig. 5).

Cold seawater (temperatures below 0 °C for most of the year) decreases the capacity of all batteries. Some instruments are powered with lithium batteries that provide a higher power density. Engineering constraints dictate the trade-offs between the various sensor battery packs and the desired sampling rates. For example, due to the large power demand of the Kongsberg Contros HydroC $pCO_2$

Microsoft Office …, 10/25/2018 12:06 PM

Microsoft Office …, 10/30/2018 10:24 AM

Microsoft Office …, 10/30/2018 10:24 AM
Moved up [2]: (Fig. 4c).

Microsoft Office …, 10/30/2018 10:24 AM
Moved (insertion) [2]

Microsoft Office …, 10/30/2018 10:24 AM

Microsoft Office …, 10/30/2018 10:24 AM

Microsoft Office …, 10/30/2018 10:23 AM

[revised manuscript text omitted]

---

## Author Response (AR3)

Dear Editor Hoppema,

Thank you very much for thoroughly reading our manuscript again. We appreciate your comments and think that it further improved our manuscript. Please find a detailed description of the changes below.

The changes are also visible in the marked-up manuscript.

Thank you again for having taken the time to further edit and improve the paper.

Best regards,
Claudine Hauri and co-authors

**Point by point response**

Throughout the manuscript, the thing that is measured is called „parameter". Please consider to use "variable" instead. A parameter should be something that is parameterized, right?
➔ "parameter" has been changed to "variable" throughout the manuscript

Sections 6 and 7 convey a very similar kind of message and for that reason they should be combined into one, i.e. in one section with the title "Concluding remarks" or "Concluding thoughts". The contents can stay as is.
➔ Sections 6 and 7 have been combined
P2, L1 „annual average temperatures" It is not clear from the text whether air temperature or sea temperature is meant. Please clarifiy.
➔ It now reads: "annual sea surface temperatures"

P3, L42-43 "Oceanographers have long struggled with finding a way to measure upper water column stratification and heat content through the fall up to the time of freeze-up in ice-covered seas." This sentence sounds as if not from a scientific work. Please delete or modify to say what the instrument is supposed to do.
➔ This was changed to:" One example of a purpose-built technology for the CEO is a novel
"freeze-up detection mooring", which measures upper water column stratification and heat content through the fall up to the time of freeze-up in ice-covered seas (Fig. 4c). It was first deployed in fall 2015."

P4, L17 I think fuller is not a correct word here, and anywhere else. Something is either full or not, but
cannot become fuller. Please change wording.
➔ This was changed to "Conversely, the CEO can help place research cruise data into a broader temporal context,…"

P6, L11 "from physics to biology" I think you are not talking about these disciplines in a strict sense. This is more about physical and biological processes, right? Please modify accordingly.

➔ This was changed to "The extensive year-round dataset derived from the CEO is providing insights into how the ecosystem operates, covering physical, chemical, and biological processes."

P6, L15 "The data and improved mechanistic understanding of the shelf ecosystem are available …" Are these really available now? I think it is too early to speak of improved mechanistic understanding.

➔ We changed it to: "The data from the observatory are available to improve biogeochemical and ecological models that allow us to test, analyze, and prepare for the future."

P6, L20-21 "Reduced sea ice may increase the potential of northward migration of subarctic species, including commercial fish species that will alter those foodwebs." This contention needs a reference.

➔ We added the following reference: Frainer, A., Primicerio, R., Kortsch, S., Aune, M., Dolgov, A.V., Fossheim, M., and Aschan, M. M.: Climate-driven changes in functional biogeography of Arctic marine fish communities, PNAS, 114, 12202-12207, https://doi.org/10.1073/pnas.1706080114, 2017.

P6, L24 "are expected" instead of "can be expected"

➔ Changed!

P6, L26-27 "Ocean acidification, warming, freshening, and de-oxygenation are large-scale issues …" I do not fully agree. Those processes also play on smaller scales, and the variation in different oceanic regions is great.

➔ Quickly arising issues such as ocean acidification, warming, freshening, and de-oxygenation require interdisciplinary efforts.

P6, L29 "do not include such a large array of disciplines" It is not clear what "such" is referring to. I think it must be deleted.

➔ This was changed to "On the other hand, continuous time-series moorings generally do not include a large array of disciplines (Newton et al., 2015)."

References: Please apply the Ocean Science format for all references. Please check to make all references uniform in format and use of abbreviations etc.
P7, L18 Deep-Sea (hyphen)
P7, L25 Add volume and pages: volume 509, pages 479–482
P7, L42 Deep-Sea (hyphen)
P8, L10 (Figure 1) at this place?
P8, L17 Deep-Sea (hyphen)
P8, L26 There is no journal mentioned.
P8, L38 Deep-Sea (hyphen)

P9, L12-13 Correct reference is: McPhee, M. G., A. Proshutinsky, J. H. Morison, M. Steele, and M. B. Alkire (2009), Rapid change in freshwater content of the Arctic Ocean, Geophys. Res. Lett., 36, L10602, doi: 10.1029/2009GL037525.

P9, L30 Deep-Sea Res.

P9, L33 Deep-Sea (hyphen)

P10, L15 Deep-Sea (hyphen)

➔ The suggested corrections and additional adjustments have been made to better align with Ocean Science's format for references

Caption Fig. 1, L9 delete: respectively

➔ Done

"The annual anomalies were computed by subtracting 0.114 °C." I do not understand why this was done. Please explain.

➔ We clarified this: "The annual anomalies were computed by subtracting the long-term mean, 15 0.114 °C."

Caption Fig.2, L6 Circulation patterns are indicated with "Current" in most cases. However, the orange arrows are called "Water". Please explain or modify.

➔ The flow patterns schematically represent our general understanding but not all regions of the 20 ocean have been bestowed with named currents (and we dont just make them up!). Some regions have had persistent currents identified and named and at these locations we use the term "current". In other instances currents have not been named and/or the most important characteristic is the type of water flowing by. In these cases we use the Water designations. We added the following explanation: ""Water" designations were used for locations, where 25 persistent currents have not been identified and named yet."

**(1) Comments from Referee 1**

Comment by D. R. Eriksen (Referee) ruth.eriksen@csiro.au

General comments: Thank you for the opportunity to review this paper. The current state of research and new approaches described are an important contribution to improving high frequency time-series in locations that are traditionally hard to access for year-round observations. The observatory described, located in a region of dynamic change, is impressive, and although this paper presents preliminary 35 analysis of results, the compilation of information on how the observatory was designed and factors that resulted in the final design are incredibly important for other research programs that use complex moorings arrays in harsh environments. The value of communicating "lessons learned" cannot be understated. I look forward to the following series of papers that provide in depth analysis of these high frequency observations and improved understanding of ecosystem dynamics in this region.

Minor comments:

P 2, l 42. Figure 4. Because this is such a great illustration of a time-series of change in a complex environment, I immediately went looking for the artists name, it took me a little while to find it. I wonder if you can highlight this more in the caption. Or provide a download link to a high -quality version that can be used with appropriate citation and acknowledgement.

P3, l 10 Do you have continuous data for estimates of MLD at the mooring location? Even a simple summary of how this changes relative to water column depth over the seasonal cycle would be useful for those of us more familiar with Antarctic cycles than Arctic cycles and ocean dynamics.

P 3, l 13- can you provide a citation for the "relatively low grazing activity"? This is an interesting point for understanding modes of carbon export compared to other polar systems. Also, any linkages to zooplankton phenology associated with both the summer and fall phytoplankton blooms.

P 3, l39 this freeze-up mooring and associated data set is fantastic. I can see wide applications.

P 4 l 12 Can you comment on how many times during a typical mooring deployment cycle you were able to obtain samples for sensor calibration purposes? The option of profiling winches is certainly attractive option, with the potential for "event based "sampling if real-time communications limitations can be overcome

P 6 l 45 SOTS observatory is described by "Eriksen et al" (not "Erikson")

**(2) Author's response**
Dear Dr. Eriksen,
	Thank you very much for taking the time to review our manuscript. We are currently working on a series of papers describing our data in more detail and are hoping to be able to submit these papers soon.
	We really appreciate your comments about the art and are hoping that it will be used widely. Besides highlighting Klara Maisch's name in the caption, we will also provide a link to a high-resolution version of her art that will be hosted on her private website, including an appropriate citation.

	We unfortunately do not have seasonal data for the mixed layer depth at the mooring location. The ice detection buoy is a first try to get a better understanding of the mixed layer depth from the time it is deployed (late July or August) until freeze up. Peralta-Ferriz and Woodgate, 2015 (Progress in Oceanography) compiled available salinity and temperature profiles from 1979-2012 and estimated a mixed layer depth minimum of 12 m in July/August and maximum of 36 m in March. Current methods to collect seasonal mixed layer depth data are too costly for our project (e.g. WHOI's bottom lander), but we are looking into other options.
	There are several studies that suggest a relatively low grazing rate for the Chukchi Sea. For example, Campbell et al 2009 (Campbell, R.G., E.B. Sherr, C.J. Ashjian, S. Plourde, B.F. Sherr, V. Hill, and D.A. Stockwell. 2009. Mesozooplankton prey preference and grazing impact in the western Arctic Ocean. *Deep Sea Research Part II: Topical Studies in Oceanography* 56 (17): 1274-1289) and

Kitamura M, Amakasu K, Kikuchi T, Nishino S (2017) Seasonal dynamics of zooplankton in the southern Chukchi Sea revealed from acoustic backscattering strength. Continental Shelf Research 133:47-58 doi:https://doi.org/10.1016/j.csr.2016.12.009.

However, co-author Catherine Lalande has been working up data from CEO's sediment trap, which contain a lot of fecal pellets, pointing towards high grazing pressure during spring. Her hypothesis is that even if there is a productive zooplankton community, primary production is so extremely high and the shelf so shallow that carbon export is considerable. These results and zooplankton phenology associated with the summer and fall blooms, including a discussion of papers reporting low grazing pressure will be presented in a manuscript that is currently in preparation.

We always take calibration samples at deployment and recovery of the observatory. Unfortunately, since the HydroC pCO2 and SeapHOx sensors need some acclimatization time (~ 2 weeks for SeapHOx), samples taken right after their deployment cannot be used for calibration. However, we are usually able to get 1 to 3 additional calibration samples in fall, depending on cruises of
opportunity and the willingness of the PIs to make the effort of collecting water samples.

All comments are directly addressed in the manuscript and described in section (3) "Authors changes in manuscript."

Thank you again for reviewing our manuscript and for your productive comments.

Best regards,
Claudine Hauri and co-authors (3) Author's changes in manuscript
P 2, l 42. Figure 4. Because this is such a great illustration of a time-series of change in a complex environment, I immediately went looking for the artists name, it took me a little while to find it. I wonder if you can highlight this more in the caption. Or provide a download link to a high -quality
version that can be used with appropriate citation and acknowledgement.

We added the following sentences to the caption of figure 3: The illustration was painted by Klara Maisch. A high-resolution version can be downloaded from her personal website at: https://klaramaisch.com/chukchi-sea-mooring-illustration.

P3, l 10 Do you have continuous data for estimates of MLD at the mooring location? Even a simple summary of how this changes relative to water column depth over the seasonal cycle would be useful for those of us more familiar with Antarctic cycles than Arctic cycles and ocean dynamics.

We added available information about the mixed layer depth to the text. These estimates are based on available data from the entire Chukchi Sea. We changed the text accordingly:
p. 3 L.1: Through heat loss, sea ice formation, and brine rejection (Fig. 3B) in late fall and winter, the water column over the Chukchi shelf becomes more saline and vertically homogenized (Weingartner et al., 2005), deepening the mixed layer depth to a maximum of ~36 m in March (Peralta-Ferriz and Woodgate, 2015).

P. 3 L. 9: During this time, the water column stratifies with inputs of fresh meltwater and heat at the surface (Fig. 3G), leading to a shoaling of the mixed layer depth to a minimum of ~12 m (Peralta-Ferriz and Woodgate, 2015). This is the time when extraordinary phytoplankton blooms occur in the nutrient rich surface waters (Fig. 3F; Hill et al., 2018).

P 3, l 13- can you provide a citation for the "relatively low grazing activity"? This is an interesting point for understanding modes of carbon export compared to other polar systems. Also, any linkages to zooplankton phenology associated with both the summer and fall phytoplankton blooms.
We modified the text to: P.3 L 12: These high rates of primary production support large fluxes of sinking particulate organic matter to the seafloor (Fig. 3I, Lalande et al., 2007), thereby sustaining a rich benthic ecosystem (Fig. 3J; Grebmeier et al., 2006, Grebmeier et al., 2015), which attracts large numbers of marine mammals that forage on the benthos (Fig. 3K; Jay et al., 2012; Hannay et al., 2013) or Arctic cod (Fig. 3L).

P 3, l39 this freeze-up mooring and associated data set is fantastic. I can see wide applications. Thank you!

P 4 l 12 Can you comment on how many times during a typical mooring deployment cycle you were able to obtain samples for sensor calibration purposes? The option of profiling winches is certainly attractive option, with the potential for "event based "sampling if real-time communications limitations can be overcome
Please see the comment above. To calibrate the 2015-2016 NO3 data record, we used two calibration samples as described in the figure 6 caption, p16, L5-8: "*In-situ* $NO_3$ water samples were collected at times of the CEO deployment and recovery, and were analysed with standard wet chemical determinations of nitrate + nitrite of frozen samples at the Chesapeake Biological Laboratory. Using the calibration samples as anchor points, a drift of 12 umol $l^{-1}$ throughout the deployment was found and corrected by linearly detrending the data."

P 6 l 45 SOTS observatory is described by "Eriksen et al" (not "Erikson")
We corrected the typo.

(1) Comments from Referee 1
Anonymous Referee #2

The authors present a blueprint for a complete observing system, capable of capturing relevant physical, chemical and biological parameters for monitoring and understanding the progress of change in a vulnerable Arctic system. While the paper doesn't really focus on new scientific results, the manuscript is a useful way to share the challenges and early successes of working with an automated observing platform in a remote region. I hope that papers with a deeper focus on the science are forthcoming. The description of the physical environment and the well-studied physical seasonality provides a nice background for the description of the observing platform and the presentation of preliminary results.

The freeze up detection method sounds great – was the 'expendable float' recovered? Would be a tough sell to leave gear like that behind by design in the Antarctic.

Minor Comments: It would have been nice to see the $CO_2$ system data that was collected once the SeapHOx sensor was added since these provide the data required to address questions relevant to the
progress of acidification which the authors outline as a key research goal. It was reassuring to see the correspondence in temperature at 34 and 43 meters depth since measurements are restricted to the subsurface – any ideas about seasonality of mixed layer depth? The $pCO_2$ supersaturations observed In Canadian waters in winter were more shallow than 34 meters (see also Shadwick et al., 2011 L&O).

(2) Author's response
Dear Referee,
        We are grateful to you for taking the time to review our manuscript, thank you. We are currently working on a series of papers describing our data in more detail and are hoping to be able to submit these papers soon. The reason we did not include pH and $pCO_2$ data in this current paper is that the
timeseries of pH and $pCO_2$ only starts in the summer of 2016. We chose to show the data return of our 2015/2016 deployment because it contained the most complete record of post-processed data to date. Similar to drifting buoys, the top float of our ice freeze up buoy is left behind. It contains the satellite communications modem, and some very basic electronics to trigger the release and a battery, which make up about 1/3 of the price.  The vast majority of the cost of the mooring is recovered, including all
scientific sensors and acoustic releases. We have put effort into trying to make them more biodegradeable by using fewer plastics and this is an ongoing process.

        We unfortunately do not have seasonal data for the mixed layer depth at the mooring location. The ice detection buoy is a first try to get a better understanding of the mixed layer depth from the time
it is deployed (late July or August) until freeze up. Peralta-Ferriz and Woodgate, 2015 (Progress in Oceanography) compiled available salinity and temperature profiles from 1979-2012 and estimated a mixed layer depth minimum of 12 m in July/August and maximum of 36 m in March. Please keep in mind that these estimates are based on available data from the entire Chukchi Sea, and may not precisely reflect the seasonality of the mixed layer depth at the observatory location. Current methods,
such as using benthic landers to collect seasonal mixed layer depth data are outside the scope of our project, but we are looking into other options. While analyzing our biogeochemical data, we will keep in mind that the mixed layer depth may at times be deeper than the location of our sensors.

(3) Author's changes in manuscript
Any ideas about seasonality of mixed layer depth?

We added available information about the mixed layer depth to the text. These estimates are based on available data from the entire Chukchi Sea. We changed the text accordingly:

p. 3 L.1: Through heat loss, sea ice formation, and brine rejection (Fig. 3B) in late fall and winter, the water column over the Chukchi shelf becomes more saline and vertically homogenized (Weingartner et al., 2005), deepening the mixed layer depth to a maximum of ~36 m in March (Peralta-Ferriz and Woodgate, 2015).

[revised manuscript text omitted]